# Tackling Insomnia Symptoms through Vestibular Stimulation in Patients with Breast Cancer: A Perspective Paper

**DOI:** 10.3390/cancers15112904

**Published:** 2023-05-25

**Authors:** Joy Perrier, Melvin Galin, Pierre Denise, Bénédicte Giffard, Gaëlle Quarck

**Affiliations:** 1Neuropsychologie et Imagerie de la Mémoire Humaine U1077, EPHE, INSERM, CHU de Caen, GIP Cyceron, PSL Université, Normandie Univ, Université de Caen Normandie, 14000 Caen, France; 2COMETE U1075, INSERM, CYCERON, CHU de Caen, Normandie Univ, Université de Caen Normandie, 14000 Caen, France

**Keywords:** sleep, insomnia, breast cancer, circadian rhythms, vestibular stimulation, chemotherapy

## Abstract

**Simple Summary:**

Patients with breast cancer frequently complaint from insomnia difficulties that can affect quality of life and cancer progression. Such difficulties may result from rest-activity (i.e., 24 h alternation of sleep and wake) rhythm alterations consistently reported in this pathology. Currently proposed approaches to counter insomnia difficulties in patients with breast cancer have positive effects only on sleep complaints and well-being. Moreover, such approaches may be difficult to implement shortly after chemotherapy. Innovatively, vestibular stimulation would be particularly suited to tackling insomnia symptoms in patients with breast cancer. Indeed, recent reports have shown that vestibular stimulation could improve rest-activity rhythm and sleep in healthy volunteers. This perspective paper aims to support the evidence of using vestibular stimulation to improve rest-activity rhythms and reduce insomnia symptoms in patients with BC, with beneficial effects on quality of life and, potentially, survival.

**Abstract:**

Insomnia symptoms are common among patients with breast cancer (BC; 20–70%) and are predictors of cancer progression and quality of life. Studies have highlighted sleep structure modifications, including increased awakenings and reduced sleep efficiency and total sleep time. Such modifications may result from circadian rhythm alterations consistently reported in this pathology and known as carcinogenic factors, including lower melatonin levels, a flattened diurnal cortisol pattern, and lower rest-activity rhythm amplitude and robustness. Cognitive behavioral therapy and physical activity are the most commonly used non-pharmacological interventions to counter insomnia difficulties in patients with BC. However, their effects on sleep structure remain unclear. Moreover, such approaches may be difficult to implement shortly after chemotherapy. Innovatively, vestibular stimulation would be particularly suited to tackling insomnia symptoms. Indeed, recent reports have shown that vestibular stimulation could resynchronize circadian rhythms and improve deep sleep in healthy volunteers. Moreover, vestibular dysfunction has been reported following chemotherapy. This perspective paper aims to support the evidence of using galvanic vestibular stimulation to resynchronize circadian rhythms and reduce insomnia symptoms in patients with BC, with beneficial effects on quality of life and, potentially, survival.

## 1. Sleep Is Often Overlooked as a Target to Improve Quality of Life and Survival in Patients with Cancer

We spend around one-third of our lives sleeping, which is critical for learning and memory, the immune system, brain energy, and plasticity [1,2]. Inadequate sleep is associated with numerous illnesses detrimental to metabolic and cardiovascular health, including a predisposition to obesity, diabetes, heart disease, and depression [3,4]. Sleep complaints, in particular, insomnia, are now well recognized in patients with non-central nervous system (CNS) cancers, not only following chemotherapy [5,6] but also before the occurrence of treatments [7,8,9]. Despite the high prevalence of complaints related to sleep disturbances in patients with cancer, such disturbances remain under-evaluated using gold-standard measures and are, thus, misunderstood in this population.

Sleep disturbances have been associated with increased risks of cancer [10,11,12] and tumor progression [13,14] in mouse models. Cancer could also indirectly influence sleep through various pathophysiological processes, including inflammation [15,16,17]. Therefore, a bi-directional relationship between sleep disturbances and inflammatory-associated cancer processes may exist. Sleep is part of one circadian rhythm (i.e., physiological processes regulated over 24 h) called the rest-activity rhythm. Previous reports have consistently shown rest-activity rhythm, and more generally, circadian rhythm, alterations in patients with cancer [18,19,20,21] that have been brought to the fore as a potential carcinogenic factor and are now considered a potential therapeutic target to improve survival in patients with cancer. Targeting not only sleep difficulties but also, more broadly, circadian rhythm alterations in patients with cancer would therefore offer the opportunity to improve their quality of life and also their survival.

## 2. Insomnia Symptoms and Sleep Structure Modifications in Patients with Breast Cancer

Insomnia is among the most common sleep complaints in patients with cancer, particularly in those with breast cancer (BC; 20–70%) compared to those with other non-CNS cancers and the general population (30%) [22,23]. Moreover, BC has become the most commonly diagnosed cancer in women worldwide [24]. Therefore, this perspective paper will focus on BC. The following section is intended to provide an overview of insomnia difficulties in patients with BC. An in-depth review of such difficulties is beyond the scope of this paper and has been specifically conducted previously [25,26,27].

Recent studies have shown that insomnia complaints are present in patients with BC even before and at diagnosis [28], before treatment initiation [25], and following treatment, mainly radiotherapy and chemotherapy [25,28,29]. Based on these results highlighting sleep complaints in patients with BC, and to tackle such difficulties, there is a need to describe the actual sleep structure modifications associated with BC and its treatments. Sleep structure disturbances in patients with BC have been mainly described using actigraphy, which indirectly measures sleep quality and quantity. However, while polysomnography (PSG) remains the gold standard for evaluating sleep quantity and quality, it has been less used in patients with cancer [30,31].

Two studies using actigraphy reported worse total sleep time and sleep efficiency before initiating chemotherapy treatments for BC [32,33] compared to normative values established in the US using actigraphy data [34]. Kreutz et al. [35] evaluated sleep parameters using actigraphy in patients with BC starting chemotherapy compared to normative data and stratified patients as good and poor sleepers. Patients did not differ from normative data, and no difference was found between good and poor sleepers. Unfortunately, no information was available about the time since the start of chemotherapy at the time of evaluation. However, the significance of these results is limited by the lack of a control group.

Two recent studies [36,37] evaluated sleep parameters using actigraphy in patients with BC at least 12 months after treatment completion (e.g., surgery, radiotherapy, or chemotherapy) or within 12 months after chemotherapy. Trivedi et al. did not find differences in sleep parameters between healthy controls and patients with BC 12 months after chemotherapy completion. Given the variability in treatments in their sample, this study does not permit us to disentangle the effects of each treatment modality. Moreover, since more than half of their sample took endocrine therapy, this could have driven their results. Ratcliff et al. found that patients with BC experienced relatively long waking after sleep onset, poor sleep efficiency, and short total sleep time compared to non-clinical norms [38]. However, no control group was included in this study.

Other studies have performed longitudinal protocols before, during, and/or after chemotherapy [39,40,41,42,43,44,45,46], of which only two included a control group without a history of cancer [42,43]. Their results showed longer total sleep time and nap time during chemotherapy compared to before treatment [39,40,44] and controls [42,43]. Conversely, Li et al. showed that chemotherapy initiation was associated with less sleep time, more arousal, and lower sleep quality compared with pretreatment and at the end of chemotherapy [46]. Beck et al. also reported a shorter sleep time on the first night after initiating chemotherapy than before and after [45]. Finally, Kuo et al. found no difference between assessments before and during chemotherapy [41]. Therefore, reports about the effects of chemotherapy on sleep parameters measured in actimetry are highly variable and do not allow us to draw a definitive conclusion. Nevertheless, a literature review published in 2015 [47] concluded that chemotherapy might accentuate sleep difficulties already present before the initiation of treatment for BC.

To our knowledge, only three studies have investigated the effects of endocrine therapy (anti-aromatase) [48,49] and radiotherapy [50] on sleep parameters measured using actimetry in patients with BC. One study showed no changes in sleep efficiency, total sleep time, and nocturnal arousals at endocrine therapy initiation compared to before [48]. This study’s lack of a control group could explain the lack of significant changes in sleep patterns with endocrine therapy. Conversely, Martin et al. [49] found that patients treated with endocrine therapy had lower sleep efficiency, more time awake, and higher activity levels at night than patients treated only with surgery and radiotherapy. Another study showed that almost half of the patients treated with radiotherapy had wakefulness and total sleep times outside the normal values [50].

Due to its ease of use, actigraphy is the primary approach used to evaluate sleep in patients with cancer. However, it does not facilitate a deep characterization of sleep and its pathophysiological modifications. In contrast, PSG is considered the gold standard for sleep evaluation. It comprises a multimodal recording (electroencephalography, electrooculography, electromyography, cardiac activity, and oximetry) and enables a deeper understanding of sleep modifications.

Previous experimental studies using PSG to evaluate sleep in patients with BC are scarce and have shown either a lack of sleep alterations following chemotherapy [51,52] or deleterious effects of chemotherapy and/or radiotherapy [53,54]. One study reported a longer deep sleep duration after radiotherapy in patients with BC than in healthy controls [53]. However, in this latest study, the delay between the PSG recording and the end of radiotherapy was unclear. In their pioneering study, Silberfab et al. [51] showed no differences between patients with BC and control subjects without a history of cancer for the primary sleep parameters of efficiency, number of awakenings, duration of stages, and sleep latency. Roscoe et al. [52] showed that patients slept more after than before chemotherapy. This result could be explained by poor sleep quality at baseline due to the stress associated with the cancer diagnosis and apprehension about chemotherapy or by an accumulation of fatigue and sleep deprivation during treatment leading to compensation after treatment.

Parker et al. compared the sleep structure of patients treated for advanced non-CNS cancers (stage 3 or 4 cancers), including 32 patients treated for BC (28% of the cohort) [54]. Their results showed that these patients had reduced sleep efficiency and amounts of the different sleep stages outside the norms established by Williams et al. in healthy participants [55]. Tag Eldin et al. [53] analyzed the sleep architecture of treated and untreated patients with BC and control subjects with no history of cancer. Patients treated with chemotherapy or radiotherapy had lower sleep time and efficiency and higher sleep latency and deep sleep duration than untreated patients and controls. Finally, the time spent in rapid eye movement (REM) sleep was lower in these patients.

These studies have shown changes in sleep structure in patients treated for cancer. Such sleep modifications may partly result from the circadian process alterations found in this population.

## 3. Circadian Rhythms Alterations in Patients with Cancer

Circadian rhythm alterations in patients with BC have been identified through physiological markers, including melatonin production, temperature, and cortisol rhythms over 24 h, and also using actigraphy. The latter is particularly suited to quantifying rest-activity rhythm (see [56] for rest-activity parameter definitions) over 24 h and is easier to use than physiological measures, although both approaches are complementary. Previous reports have indicated lower melatonin levels before and during chemotherapy administration [46]. Circadian disruptions such as hot flushes (i.e., a sudden wave of mild or intense body heat caused by rapid hormonal changes) [57] and flattened diurnal cortisol patterns [58,59] have been reported, supporting the view of circadian rhythm alterations in patients with BC. Using actigraphy, a lower amplitude, mean activity, and robustness of the rest-activity rhythm were identified in patients with BC during chemotherapy [42,46,60]. For example, Li et al. [46] showed that the first administration of chemotherapy was associated with an altered rest-activity rhythm, with a decrease in its amplitude, mean activity, and robustness compared with pretreatment. These changes appeared to attenuate as the treatment cycles were administered. Conversely, Sultan et al. [61] showed a worsening of alterations in the rest-activity rhythm, with a decrease in mean activity and amplitude and a shift in peak activity over chemotherapy cycles. However, the lack of measurements before chemotherapy initiation leaves open the extent to which these alterations existed before chemotherapy.

In a longitudinal study, Liu et al. found alterations in rest-activity rhythm in patients with BC compared with control subjects before and during chemotherapy [40]. Before and after chemotherapy, the patients’ rhythm showed reduced amplitude, mean activity, and robustness compared to controls. In patients, chemotherapy treatment was also associated with decreased amplitude and mean activity of the rest-activity rhythm compared to before chemotherapy initiation. In addition, two recent studies based on the same cohort showed that changes in rest-activity rhythm associated with chemotherapy, reflected by its decreased amplitude, mean activity, and intradaily variability, appeared to persist for up to five years after BC diagnosis [62,63]. In these two studies, the mean diurnal activity was lower, and its intradaily variability was larger in patients with BC than in healthy controls, suggesting an alteration in the rest-activity rhythm over 24 h in patients, even at a distance from their treatment. Overall, these results suggest that chemotherapy negatively impacts circadian rhythms, notably on the rest-activity rhythm parameters, which appear to persist years after chemotherapy. More recently, our team has shown that the amplitude of the rest-activity rhythm was reduced in patients with BC not treated with chemotherapy compared to healthy controls beyond the effects of endocrine therapy [49]. These results suggest that rest-activity modifications may be altered even before the initiation of adjuvant treatments and could be further exacerbated by chemotherapy. These results are summarized in Table 1.

Rest-activity rhythm modifications might be partly responsible for sleep disturbances in patients with BC. Indeed, associations have been shown between greater fatigue and altered circadian rhythms [64] but with greater time spent in bed [65] in nap periods [39]. These results suggest that fatigue would lead to an adaptation of sleep time over 24 h. Increased fatigue following cancer and its treatment could therefore induce alterations in the rest-activity rhythm and poor sleep habits (i.e., less activity time and more napping time), leading to the development or persistence of sleep disturbances. Moreover, circadian rhythm alterations have been associated with lower survival, calling for further studies to evaluate and address these modifications to improve sleep, quality of life, and also survival.

## 4. Associations with Survival

The hypothesis that activity-rest rhythm disorders could be carcinogenic (i.e., contribute to the occurrence of cancer) has emerged due to epidemiological studies showing an increased risk of developing BC in individuals performing shift work. It has been proposed that nocturnal exposure to light would suppress melatonin (a key regulator of central and peripheral oscillators) secretion by the pineal gland [12,66,67]. The model developed by Sephton et al. [68,69] has proposed a more precise interconnection about potential pathways through which circadian dysregulation could mediate psychosocial effects on cancer progression. They proposed that altered circadian rhythms and anxiety-depressive factors might contribute to tumor growth by deregulating glucocorticoid secretion by the hypothalamic–pituitary–adrenal axis [70]. In support of this model, a cortisol spike has been observed in patients with advanced BC compared to healthy controls. Such a higher nighttime peak was associated with a poorer prognosis, including more rapid development of metastases [71].

It has also been proposed that altered sleep may be one modifiable factor contributing to breast oncogenesis [10,11,12]. For example, a recent study on patients with advanced BC found that better sleep quality and fewer arousals, measured by actimetry, were associated with significantly reduced mortality risk over six years of follow-up [72]. Sleep disorders appear to be associated with an increased risk and/or higher aggressiveness of cancer [73,74]. However, our understanding of the underlying mechanisms needs further development and requires a multidisciplinary approach.

Screening for sleep disorders could be performed more systematically in patients with BC in the context of research protocols or clinical practice when they complain of difficulties in their daytime functioning. This screening would allow us to better understand to what extent sleep disorders influence the occurrence, aggressiveness, and recovery of BC and non-cerebral cancers in general and to treat sleep pathologies that potentially impact patients’ quality of life.

## 5. Perspective of Using Vestibular Stimulation to Improve Sleep in Patients with BC

Commonly prescribed drugs such as benzodiazepines have well-known adverse side effects [75,76,77]. Therefore, non-pharmacological interventions have gained increasing attention as an alternative first-line approach in recent years. Current non-drug approaches have been tested and partly validated to improve the quality of life and sleep complaints of patients with BC during and after treatments [78]. Such therapies include adapted physical activity or cognitive behavioral therapy that appear to positively affect quality of life, self-esteem [79,80,81,82], and subjective reports of insomnia symptoms even in the long term [83,84,85,86,87,88]. However, previous studies have provided a low quality of evidence [89].

However, while positively affecting well-being and self-reported sleep difficulties, these approaches are not explicitly targeting potential carcinogenic modifiable factors such as circadian rhythms and sleep. A challenge also remains when making non-drug therapies available and easily accessible to patients to improve adherence [90,91], highlighting the need for other approaches, such as vestibular stimulation. Indeed, previous reports suggest that chemotherapy for BC may affect vestibular function [92,93,94,95]. Moreover, considering the implication of the vestibular system in circadian regulation, its stimulation may offer a unique opportunity to regulate circadian rhythms and improve sleep in patients with BC.

The vestibular system is located within the inner ear next to the cochlear organ. It comprises three semi-circular canals detecting three-dimensional angular head velocity and two otolithic sensors detecting linear acceleration.

Besides its known role in detecting head movements and orientation, promising findings from lesion studies have also suggested that the vestibular system could constitute an input to the circadian clock or be involved in circadian regulation and synchronization [96,97,98]. Fuller et al. reported an association between the vestibular system and the circadian pacemaker based on animal studies [99,100,101]. In support of this hypothesis, bilateral vestibular lesions in rats led to a sharp fall in their core temperature and a disruption in their daily rhythmicity [96]. Similarly, a study in healthy volunteers reported a significant phase advance in the rest-activity rhythm using vestibular stimulation through a rotary chair at 18:00 compared to sham stimulation (i.e., lack of chair inclination and movement) [102]. These results support the hypothesis of the involvement of vestibular inputs in rest-activity rhythm regulation.

Since sleep is intimately associated with the rest-activity rhythm, these results suggested a potential association between sleep and the vestibular system. Patients with bilateral vestibular loss had abnormal sleep patterns and shorter sleep duration than healthy controls [97]. A recent study used actigraphy to quantify sleep in patients with unilateral vestibular hypofunction, finding that they slept less and took longer to fall asleep than healthy controls [103]. Recent epidemiological evidence has also shown abnormal sleep duration in patients with vestibular vertigo [104]. In addition, several studies have highlighted a potential association between sleep apnea (i.e., one of the most frequent sleep disorders) and vestibular impairments. However, this association remains to be clearly established [105,106].

The putative causal link between the vestibular system and sleep is strengthened by recent reports showing that the rehabilitation or stimulation of the vestibular system could promote sleep. In-hospital vestibular rehabilitation for chronic dizziness improved sleep complaints compared to before therapy [107]. Participants were taught to perform the rehabilitation program for 30 min four times a day over five days when they were in the hospital, and this program was conducted in groups. Similarly, compared to a stationary condition, continuous rocking (at 0.25 Hz) during an afternoon nap or the night promoted sleep by reducing latency into and maintenance of deep sleep (non-REM) in healthy volunteers [108,109]. Finally, a study evaluating the effects of a recliner chair with a rocking motion on sleep in healthy volunteers reported a decrease in light sleep and an increase in deep sleep when the chair moved compared to being stationary [110].

Positive effects of vestibular stimulation appear to be mediated by a decrease in the arousal level resulting from cholinergic tonus modulation and rhythmic entrainment of thalamocortical activity [111]. Older adults are more prone to experience sleep difficulties and are therefore of particular interest in using vestibular stimulation as a sleep-promoting intervention. However, three recent studies did not report sleep improvements using night or afternoon nap vestibular stimulation through rocking [112,113,114]. While this lack of significant effects of rocking on sleep may be due to the already high sleep efficiency of the participants, it could also be argued that the stimulation protocol was inadequate for efficiently stimulating the vestibular system.

Altogether, these results partly support (1) the involvement of the vestibular system as an input to circadian regulation and (2) the positive effects of vestibular rehabilitation and stimulation on sleep. The mechanisms involved remain to be understood, which will be required to form a consensus about such beneficial effects. Indeed, it might be argued that sleep and circadian disorders are not specific to vestibular lesions. Therefore, there is a need to highlight functional associations between the vestibular system and cortical/sub-cortical regions involved in sleep regulation. The association between the vestibular system and circadian rhythms is currently supported by neuro-anatomical pathways between the median vestibular nuclei and the suprachiasmatic nucleus (i.e., circadian pacemaker) [98,115].

Considering that patients with BC could have both circadian rhythm and sleep difficulties that could negatively influence their recovery after chemotherapy, innovative approaches targeting both phenomena are needed. Galvanic vestibular stimulation (GVS) could be particularly well suited for this purpose [116]. Indeed, GVS is based on stimulation of the peripheral vestibular organ via direct activation of vestibular nerve afferent fibers at the spike trigger zone, bypassing hair cell synapses [117,118] using electrical stimulation at the back of each ear. The question of which parts of the vestibule GVS activates has been debated over the past decade. A recent review proposed that GVS’s effects would result from a central semi-circular canal-otolith signal convergence and integration, further converging in central vestibular neurons [119]. Based on previous literature, working hypotheses can be proposed for GVS’s specific effects on circadian rhythms and sleep. Notably, the orexinergic system shares a bi-directional relationship with the vestibular system [119] and could mediate these effects. A functional hypothesis is that the vestibular system monitors the daily motion quantity and informs the orexinergic neurons, influencing the sleep–wake state switch [98].

Altogether, these previous reports support using GVS to limit circadian rhythms and sleep disturbances in patients with BC. Compared to other non-drug approaches currently available, GVS has the additional benefit of offsetting the decrease in vestibular stimulation resulting from hair cell impairments caused by chemotherapy agents.

## 6. Conclusions and Research Agenda

This perspective paper has highlighted sleep and circadian rhythm disturbances in patients with BC before and after chemotherapy. Several studies also suggest sleep difficulties and circadian rhythm alterations due to chemotherapy effects. Moreover, results from previous studies have repeatedly shown that sleep difficulties are associated with a lower quality of life and that circadian rhythm alterations and, to a lesser extent, sleep difficulties are associated with lower survival. These results call for approaches targeting circadian rhythm alterations and sleep difficulties that could be implemented shortly after chemotherapy in patients with BC.

Given the beneficial effects of vestibular stimulation on circadian rhythms and sleep and knowing the potential vestibulotoxic effects of chemotherapy, we have proposed using vestibular stimulation as a new non-pharmacological intervention. Vestibular stimulation is expected to specifically reduce circadian rhythm alterations and sleep difficulties in patients with BC after chemotherapy. GVS would be of particular interest since it has three potential benefits. Firstly, it allows remote vestibular stimulation to be performed at home. Patients with BC may have difficulties attending oncological support due to increased fatigue or distance between home and the hospital. Therefore, having a remote approach is of interest to increase adherence. Secondly, it is a safe and easy-to-use approach. Several systems can currently be used to pre-program stimulation and avoid patients doing so, avoiding accidental mishandling. Thirdly, videoconferencing during the stimulation could be considered to ensure social support and increase patient adherence.

Overall, this perspective paper supports using vestibular stimulation, particularly GVS, in patients with BC to resynchronize their circadian rhythms and improve their sleep, benefiting their quality of life and, potentially, survival.

## 7. Research Agenda

In laboratory settings:Quantify the beneficial effects of GVS on circadian rhythms and sleep in patients with BC after chemotherapy as a proof of concept.Compare the quality-of-life outcomes between patients using GVS and those not using stimulation.Compare the survival outcomes between patients using GVS and those not using stimulation.Determine the best time to use GVS in patients (e.g., before, during, or after [and how long after] chemotherapy).Determine and quantify such beneficial effects in other cancer populations.Explore the physiological and neurofunctional correlates of GVS to clarify its underlying mechanisms.

The above steps will further facilitate the implementation of GVS in oncological support in close partnership with clinical oncologists.

This research agenda requires close collaboration between several scientific disciplines (i.e., neuroscientists, neuropsychologists, and neurophysiologists) and clinicians that frequently work in different departments and speak different scientific languages. Moreover, this topic would benefit from translational studies that are currently rarely conducted, mainly for the same reasons.

## Figures and Tables

**Table 1 cancers-15-02904-t001:** The main results related to circadian rhythm modifications in patients with breast cancer (BC), including melatonin, hot flushes, cortisol, and actigraphy-derived (rest-activity rhythm) parameters.

Authors, Year	Measures	N	Groups	Cancer Stages	Treatments	Time of Assessments	Main Results
Li et al., 2019 [46]	Urinary melatonin	180	BC	I–III	Surgery and CT	Before and during first and last cycles of CT	Mean levels of melatonin *Before CT < after CT*
Carpenter et al., 2004 [57]	Hot flashes using skin conductance	21	BC	0–III	Surgery and RT and/or CT	After treatments, >5 years post-diagnosis	High variability in the rhythm of hot flashes
Abercrombie et al., 2004 [58,59]	Salivary cortisol	17 31	BC HC	Metastatic	NA	NA	Rhythm over 24 h *BC < HC*
Hsiao et al., 2017 [58,59]	Salivary cortisol	85	BC	0–III	Surgery and RT and/or CT	Before (T0) and after treatments (T0 + 2 months; T0 + 5 months; T0 + 8 months; T0 + 14 months)	Flatter cortisol levels tend to recover *over the course of treatments*
Ancoli-Israël et al., 2014 [42,46,60]	Actigraphy (3 h)	68	BC HC	I–III	Surgery and RT and/or CT	Before CT and at Cycle 4 of CT and 1 year post-CT	Naps time before CT *BC > HC* Robustness * *BC during Cycle 4 < before CT and HC* Naps time and robustness 1 year post-CT *BC = HC*
Li et al., 2019 [42,46,60]	Actigraphy (7 days)	180	BC	I–III	Surgery and CT	Before and during first and last cycles of CT	Amplitude, mean activity, and robustness * *Before CT > after CT*
Savard et al., 2009 [42,46,60]	Actigraphy (3 days)	95	BC	I–III	Surgery and CT + ET (27% of the sample)	Before CT and at Cycle 1 and 4 of CT	Amplitude, mean activity, and robustness * *First week of each CT cycle < before CT*
Roveda et al., 2019 [62]	Actigraphy (7 days)	15 13	BC HC	NA	NA	5 years post-diagnosis	Mean activity and amplitude *BC < HC*
Sultan et al., 2016 [62]	Actigraphy (3–4 days)	25	BC	II	CT	Cycles 1, 3, and 6 of CT	Mean activity and amplitude decreased *over the course of CT* Shift in peak activity *over the course of CT*
Liu et al., 2013 [40]	Actigraphy (3 days)	79 61	BC HC	I–III	Surgery or/and CT	Before CT and at the end of Cycle 4 of CT	Amplitude and mean activity *Before CT > Cycle 4 of CT* Amplitude, robustness *, mean activity *Both before and after Cycle 4 of CT, BC < HC*
Galasso et al., 2019 [63]	Actigraphy (7 days)	15 13	BC HC	NA	NA	5 years post-diagnosis	Intradaily variability * *BC > HC*
Martin et al., 2021 [49]	Actigraphy (15 days)	18 18 16	BC with ET BC without ET HC	0–II	Surgery and RT	6 months post-RT	Amplitude *BC < HC* Interdaily stability * *ET− < HC*

Note: N, number of participants in each group; BC, patients with breast cancer; HC, healthy controls; NA, unavailable; CT, chemotherapy; RT, radiotherapy; ET−, with/without endocrine therapy; *, the robustness of the rest-activity rhythm refers to the relative amplitude that is the relative difference between the 10 h with maximal activity and the 5 h with minimal activity. The interdaily stability is the ratio between the variance of the average 24 h pattern around the mean and the overall variance, with higher values indicative of rhythm stability. The intradaily variability is calculated as the ratio of the mean squares of the difference between successive hours and the mean squares around the grand mean. Comparisons are given in italic for ease of reading. See [56] for rest-activity measure definitions.

## Data Availability

Not applicable.

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
