# Peer review of "Tackling Insomnia Symptoms through Vestibular Stimulation in Patients with Breast Cancer: A Perspective Paper"

_cancers, 2023, doi:10.3390/cancers15112904_

Round 1

Reviewer 1 Report

I would like to congratulate the authors for the ideea of this article! Sleep difficulties are common in cancer Pacients and Little investigated and treated by oncologists. The help of psychologists is welcome.
 I think that the conclusion secțion should be longer and also include the possible ideea of new type of  non pharmacologycal intervention that may be investigated in the future. That will be very usefull for the readers.

Sincerely yours,

Author Response

Dear Reviewers,

Thank you for taking time for reviewing our manuscript and providing us with feedback. Comments from the reviewers have surely helped to improve the clarity of the manuscript.

Please find below a point-by-point explanation of how we have addressed comments, which we hope you will find satisfactory. Please note that the page and line numbers, indicated in italic, refer to the track changes version of the resubmitted manuscript.

REVIEWER 1
I would like to congratulate the authors for the idea of this article! Sleep difficulties are common in cancer patients and little investigated and treated by oncologists. The help of psychologists is welcome.
 I think that the conclusion section should be longer and also include the possible idea of new type of non-pharmacological intervention that may be investigated in the future. That will be very useful for the readers.

Sincerely yours,

Answer: We thank the reviewer for his/her kind words regarding our manuscript. We have modified and extended the conclusion and we put more emphasis on the approach we suggested, namely galvanic vestibular stimulation. Moreover, a research agenda has been added. See Pages 16-18.

Reviewer 2 Report

The abstract needs quantification. sleep gap analysis is not done properly. The section 1 and 2 cited with more than 60 references. The table representation will be provided for section 3. Too many points on discussion. Kindly reduce to your objective specific only.

Author Response

Dear Reviewers,

Thank you for taking time for reviewing our manuscript and providing us with feedback. Comments from the reviewers have surely helped to improve the clarity of the manuscript.

Please find below a point-by-point explanation of how we have addressed comments, which we hope you will find satisfactory. Please note that the page and line numbers, indicated in italic, refer to the track changes version of the resubmitted manuscript.

REVIEWER 2

We thank the reviewer for his/her comments to improve the content and the clarity of our manuscript and we hope that modifications adequately addressed these comments.

  • The abstract needs quantification.

Answer: In order to add quantification in the abstract, we added the percentages of sleep complaints as well as the specific sleep structure and circadian rhythms modifications that have been highlighted in breast cancer patients. We hope these modifications adequately addressed reviewer’s comment.

  • Sleep gap analysis is not done properly.

Answer: We agree that, in this manuscript, we focused only on insomnia difficulties experienced by breast cancer patients while other sleep disorders may also exist in this population. We thus modified the title of the article and of Section II to be clearer about the fact that the manuscript was centered around insomnia difficulties in breast cancer patients. The purpose of this section was more to introduce insomnia difficulties in breast cancer patients rather than making a systematic review about sleep difficulties in these patients. We also added a sentence in that sense “The following section is intended to give an overview of insomnia difficulties in BC patients. A deep review about such difficulties is beyond the scope of this paper and has been specifically done previously [25–27].See Page 4, lines 7-9.

  • The section 1 and 2 cited with more than 60 references.

Answer: We thank the reviewer for this comment and we understood that too much references were cited in both sections. We have reduced the text and the associated references in Section 2. There are now around 45-50 references related to insomnia and sleep macrostructure modifications in breast cancer patients in both sections. We hope this reduction addressed reviewer’s comment. We have reduced some parts of this section (Page 4, lines 20-24) and we have removed the sentence Page 6, lines 17-19 and the last paragraph of this section in order to only focus on sleep macrostructure modifications that are related to the aim of the manuscript. See Page 6, lines 17-19 and Pages 7-8.

We hope that such modifications will lead to a clearer overview of insomnia difficulties in breast cancer patients in order to further introduce the need to take care of such difficulties.

  • The table representation will be provided for section 3.

Answer: A table now summarize results from Section 3. See Table 1.

  • Too many points on discussion. Kindly reduce to your objective specific only.

Answer: We have reduced Section 5, especially by removing the part related to balance deficits following chemotherapy. We hope that such modifications have improved the clarity of this section. See Pages 11-14.

Round 2

Reviewer 2 Report

All the corrections are included in the paper and hence, there is no need for further review from our side.